# Subfield-specific interneuron circuits govern the hippocampal response to novelty in male mice

Thomas Hainmueller [1,2,3,4] ✉, Aurore Cazala[1,4], Li-Wen Huang[1] & Marlene Bartos [1] ✉

The hippocampus is the brain's center for episodic memories. Its subregions, the dentate gyrus and CA1-3, are differentially involved in memory encoding and recall. Hippocampal principal cells represent episodic features like movement, space, and context, but less is known about GABAergic interneurons. Here, we performed two-photon calcium imaging of parvalbumin- and somatostatin-expressing interneurons in the dentate gyrus and CA1-3 of male mice exploring virtual environments. Parvalbumin-interneurons increased activity with running-speed and reduced it in novel environments. Somatostatin-interneurons in CA1-3 behaved similar to parvalbumin-expressing cells, but their dentate gyrus counterparts increased activity during rest and in novel environments. Congruently, chemogenetic silencing of dentate parvalbumin-interneurons had prominent effects in familiar contexts, while silencing somatostatin-expressing cells increased similarity of granule cell representations between novel and familiar environments. Our data indicate unique roles for parvalbumin- and somatostatin-positive interneurons in the dentate gyrus that are distinct from those in CA1-3 and may support routing of novel information.

The hippocampus is critical for autobiographic memory, spatial navigation, and novelty-detection, among others. Individual hippocampal subfields contribute differentially to these functions[1-4], and show differences in their cellular composition, afferent input pathways and microcircuit connectivity[5,6]. The dentate gyrus (DG) is crucial for discriminating novel contents from similar familiar ones[2,3,6,7], which is supported by DG granule cells (GCs) with their particularly low excitability, the absence of recurrent synaptic connections, and their low divergence towards CA3 principal cells[8]. In contrast, CA3 aids fast encoding of novel information[1], which is thought to be supported by rapidly inducible synaptic plasticity in a strongly interconnected recurrent principal cell network[5]. In line with this idea, CA1-3 principal cells are highly recruited upon first exposure to a novel scenery[9-11], while activation of DG GCs is reduced[9,10] (but see ref. 12). GABAergic

interneurons (INs) may regulate this differential response to novelty, but their influence on the representation of novel environments by neuronal populations has not yet been studied in most hippocampal subfields.

While GABAergic INs make up for only 10–15% of all hippocampal neurons[13-15], they form a heterogeneous group distinguished by genetic, molecular, morphological and physiological features[13,15-18]. Hippocampal INs are broadly categorized into two main classes defined by the subcellular compartments targeted by their output synapses[13,16,19]: Perisomatic-inhibitory INs, which mostly express the calcium-binding protein parvalbumin (PV)[13,20], and dendritic-inhibitory INs, many of which express the neuropeptide somatostatin (SOM)[16,21,22]. Recordings in brain slices unveiled some of their functional principles: First, they jointly regulate the activity of hippocampal

[1]Institute for Physiology I, University of Freiburg, Medical Faculty, 79104 Freiburg, Germany. [2]NYU Neuroscience Institute, 435 East 30th Street, New York, NY 10016, USA. [3]Department of Psychiatry, New York University Langone Medical Center, New York, NY 10016, USA. [4]These authors contributed equally: Thomas Hainmueller, Aurore Cazala. ✉e-mail: Thomas.Hainmueller@nyulangone.org; Marlene.Bartos@physiologie.uni-freiburg.de

principal cells by feedforward and feedback inhibition[23,24]. Second, their glutamatergic inputs undergo long-term plasticity, which alters their recruitment[25–28]. Third, they exhibit efficient inhibitory control of specific subcellular compartments[20,29–31] and contribute to fast network oscillations[32,33]. However, information on behavior-related activity dynamics of hippocampal PV- and SOM-cells is scarce.

Recent studies recorded the activity of neurochemically defined GABAergic cell types in CA1 and CA3 during explorative behavior[24,34–41]. They observed a reduction of CA1 SOM-IN activity in novel environments, but had conflicting results regarding CA1 PV-INs[37–39]. In vivo recordings from DG INs are very limited and were obtained from few unidentified INs[9,42,43]. One recent study has characterized basic firing rates of identified DG SOM-INs[44,45] and we have previously attempted to identify DG PV-IN in densely labeled hippocampal neuron populations[10]. However, the relationship of their activity to behavioral and cognitive variables remains unknown. We therefore selectively labeled and recorded calcium activity of PV- and SOM-INs in the hippocampal DG, CA2/3 and CA1 area in behaving mice using two-photon imaging. We further chemogenetically manipulated the activities of SOM- and PV-INs in the DG and CA1, and compared their impact on

principal cell activity in the respective subfields. We find different activity patterns and network impact of DG SOM- and PV-INs compared to their CA2/3 and CA1 counterparts in several behavior- and memory-associated domains. We conclude that INs may thereby support the unique computational demands of their respective hippocampal subfields.

## Results

### Head-fixed two-photon calcium imaging of hippocampal INs

To record hippocampal IN activity during behavior, we trained mice to run on a spherical treadmill while head-fixed under a two-photon microscope (Fig. 1a). A virtual environment, consisting of a 4-meter long circular track, was displayed on monitors surrounding the animal[10] (Fig. 1b). Before commencing the imaging experiments, mice were familiarized to running on one circular track (familiar environment) for at least ten days. Then, imaging experiments started and mice were alternately exposed to the familiar and a visually distinct, novel track, selected from a pool of different tracks (Fig. 1b; Methods).

To record IN activity, we injected SOM-Cre and PV-Cre mice with adeno-associated viruses (AAVs) containing a floxed copy of the green-

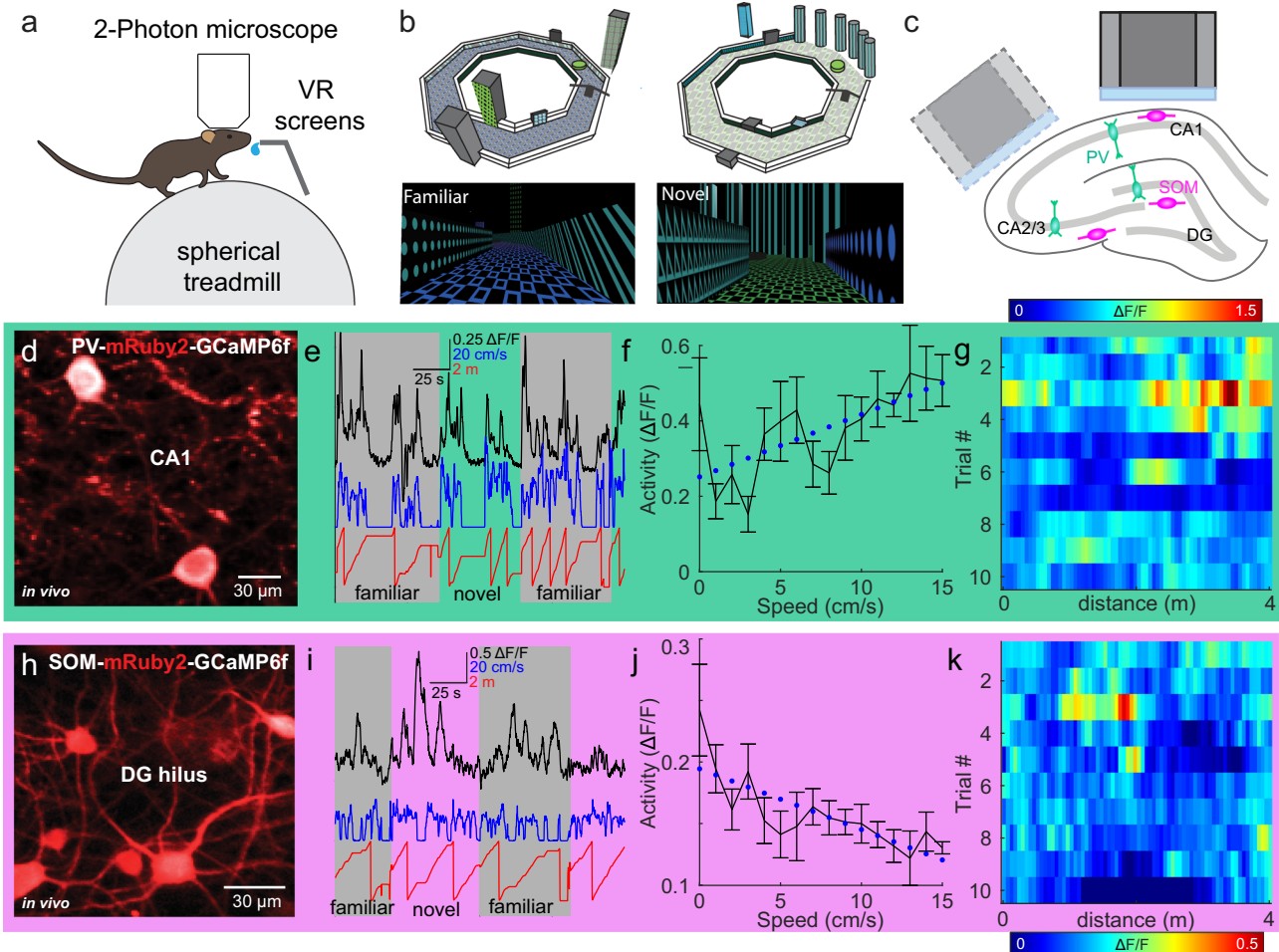

**Fig. 1 | Illustration of the recording conditions for hippocampal interneuron activity in head-fixed mice navigating through virtual environments.**
**a** Schematic of the recording setup. **b** Virtual circular tracks and behavioral layout. Familiar and novel tracks are characterized by different wall and floor patterns. **c** Schematic of the implantation. The lateral window implantation for CA2/3 recordings (Methods) is depicted with dashed lines. **d** Illustrative, time-averaged GCaMP6f (white) and mRuby2 (red) fluorescence of CA1 PV-cells in vivo, representative for the recording conditions in $n = 12$ experiments. **e** Fluorescence signal (black), animal running speed (blue) and position on the circular track (red) of an

illustrative example CA1 PV-interneuron (IN) over time. **f** Calcium activity of a PV-IN ($n = 1$, same as in **d**) as a function of running speed. Dots with lines represent mean ± SEM; dotted blue lines represent linear fit to the data with an offset. **g** Activity (color-coded) of the same IN as in **e**, **f** as a function of position on the familiar circular track over multiple trials, consisting of 60 s running on the respective tracks. **h**–**k** Same as in **d**–**g**, but for an example SOM-IN ($n = 1$) in the DG. Image in h is representative for $n = 8$ experiments. For cell examples from all recorded areas, see Supplementary Fig. 1d–i.

fluorescent calcium indicator GCaMP6f fused with the red-fluorescent protein mRuby2 (AAV1.CAG.FLEX.mRuby2.GSG.P2A.GCaMP6f.WPRE. pA) to selectively express both markers in SOM- or PV-INs (Fig. 1d, h; Supplementary Fig. 1a, b). Next, we implanted a transcortical imaging window above the external capsule[10], to image INs in the dorsal DG and CA1 at depths of ~700 μm and ~150 μm, respectively (Fig. 1c, d, h; Supplementary Fig. 1c). To image the dorsal CA3, we implanted a more lateral window and tilted the microscope objective by 20° (Fig. 1c; Methods). Thereby, most of the recorded INs were located within CA3 but some CA2 INs may also have been included[10,46]. We therefore refer to these recordings as CA2/3. We used fast volumetric resonant scanning (30 Hz; Methods) to simultaneously record from 14.13 (range 3–73) INs per session (see Supplementary Table 1). Given the uneven distribution of cells across recording sessions and animals and the substantial variance of responses between individual cells, which tends to be larger than that between animals or sessions (Supplementary Fig. 2), we used cells as the primary unit of statistical comparison throughout the manuscript. In total, we recorded 133, 109, and 306 PV-INs and 119, 237, and 113 SOM-INs in the dorsal DG, CA2/3 and CA1, respectively. Somata of PV-cells were imaged in the granule cell and pyramidal cell layer or at the borders to the neighboring dendritic layers. Somata of SOM-cells were located in the hilus of the DG and the *stratum oriens* of CA1-3. Due to their high discharge rates[44,47], hippocampal INs often lack discernible calcium transients commonly observed in principal cells[10,48–50]. Nevertheless, we observed modulation of GCaMP6f-mediated fluorescence by behavioral variables, such as the animal's running speed and the position in the virtual environment in several of the recorded INs (Fig. 1d–k).

## Differential speed modulation of DG and hippocampal SOM-INs

Activity of cortical and CA1 PV-INs is modulated by locomotion speed[24,35,36,39,51]. While most CA1 PV- and SOM-INs show increasing activity with higher running speeds, in line with their strong speed-modulated septal inputs[52,53], recent studies observed a small but significant population of negatively speed-modulated INs[36,39,54]. Congruently, we found that the activity of 63.7% of PV-INs and 56.6% of SOM-INs in CA1 had a significant positive correlation with the animal's running speed while only 3.9% and 5.3% of cells, respectively, showed significant negative speed-modulation (Fig. 2a, b, *upper row*). A similar distribution was observed for CA2/3 PV- and SOM-INs (Fig. 2a, b, *middle row*). Intriguingly, this distribution was significantly different in the DG ($\chi^2$ with 10 degrees of freedom = 141.56; $p = 2.02*10^{-25}$), where ≤50% of the INs showed significant speed-modulation (Fig. 2a, b, *lower row*). Particularly, more DG SOM-INs were negatively (26.1%), rather than positively (17.6%) modulated by running speed. This feature was also reflected in a significantly lower slope of speed modulation of DG SOM-INs compared to CA1-3 SOM-INs and PV-INs throughout the hippocampus (Fig. 2c; Supplementary Fig. 2a). Furthermore, many DG SOM-cells showed higher activity during phases of immobility or very slow movement (<2 cm/s; Methods) than during running (Fig. 1i, j; Supplementary Fig. 1i). To quantify this observation, we divided the mean activity rate during moving periods by that during immobility and very slow movement. This ratio was significantly smaller for DG SOM-INs than for SOM- and PV-INs in all other hippocampal areas (Fig. 2d). However, this observation was only significant in comparison to CA1 PV-cells when data were aggregated by session (Supplementary Fig. 2b; Supplementary Table 2). Notably, DG SOM-INs were also the only of the observed IN populations that did not show a significant activity increase from immobility to moving periods (Fig. 2d). Running speed profiles were highly similar between mouse lines (Supplementary Fig. 3a) and there was no difference in the mean running speeds and the number of laps per session (Supplementary Fig. 3b, c). Finally, running speed was not different between virtual environments (Supplementary Fig. 3d, f, g), indicating that differences in running-speed modulation were not an artifact of behavioral differences between the

mouse lines or the behavioral contexts. Thus, our data demonstrate that most PV- and SOM-INs in CA1-3 are positively modulated by locomotion while a large fraction of SOM-INs in the DG is negatively speed modulated with highest activity during periods of little or no movement.

## Spatial selectivity of DG and CA1-3 PV- and SOM-INs

Hippocampal principal cells form an internal representation of space, in which individual neurons are selectively active in distinct locations of the environment. CA1-INs occasionally show spatial selectivity[54–57] and a recent study observed small fractions of spatially tuned INs in a variety of genetically defined CA1 IN types[39]. While the spatial resolution of calcium signals in our recordings is inherently limited by factors such as the acquisition rate and decay time constant of GCaMP, several of the recorded neurons appeared to show at least some degree of spatial modulation in their activity response profiles (Supplementary Fig. 4a). To analyze the degree of spatial tuning of IN activity (Fig. 3a), we first calculated the spatial information (SI) content per IN and compared it to bootstrap data obtained by circularly shuffling the animal's position and calcium activity of the same neuron (Methods). This analysis revealed significant SI in a small, but consistent fraction of about 7–13% of PV- and SOM-INs in all hippocampal subfields (Fig. 3b). This fraction did not differ significantly between PV- and SOM-INs or any of the hippocampal subfields ($\chi^2$ with 6 degrees of freedom = 7.08; $p = 0.215$).

To test whether this finding was robust on the population level, we next compared INs of each type and region, group by group, against matched bootstrap values from the same population. SI was modestly above chance for SOM-IN populations in CA1 and CA2/3 and for PV-IN populations anywhere except in CA2/3 ($p = 0.081$; Fig. 3c; Supplementary Fig. 2c). We also tested an alternative measure of positional information that considers discrete levels of activity as a means of conveying information about the animal's location[58]. We found that information by this measure was consistently above chance levels (Supplementary Fig. 4c). Similarly, spatial coherence as a measure for the local smoothness of the spatial tuning curve (Methods), was above chance for all IN groups (Fig. 3d), although that observation was not consistent in CA3 when data were aggregated per session or per animal (Supplementary Fig. 2d). Analysis of spatial vector-tuning (Supplementary Fig. 4a, b) further confirmed our findings by showing that tuning was above bootstrap-levels in all IN populations. Finally, we analyzed the stability of spatial activity on the familiar track between runs in the first- and second half of the recording session. Like spatial coherence, within-session stability was above chance levels for all IN types and hippocampal areas (Fig. 3e; Supplementary Fig. 2e), confirming low, but consistent, spatial tuning. However, mean values for spatial tuning parameters (SI, vectorial tuning, trial-by-trial reliability and within-session stability) in all IN groups were dwarfed by those of CA1 principal cells recorded under similar experimental conditions[10] (Supplementary Figs. 4d, e, g and 5a–c). The only exception from this pattern was positional information[58] for which levels of CA1 PV-INs exceeded those of pyramidal cells (Supplementary Fig. 4f). Similar results were obtained when comparing only cells with significant SI, indicating that even spatially tuned INs do not reach the accuracy of hippocampal principal cells (Supplementary Fig. 5d-f). Thus, while we observed spatial tuning in hippocampal INs of all areas and types, it was relatively weak compared to principal cells by almost all measures and present only in a small fraction of PV- and SOM-INs.

## Differential response to novelty of DG and CA1-3 PV- and SOM-INs

Hippocampal subfields show characteristic responses when animals transition to novel environments: Principal cell activity increases in CA1 and CA2/3, but declines in the DG[9,10]. To probe the role of PV- and

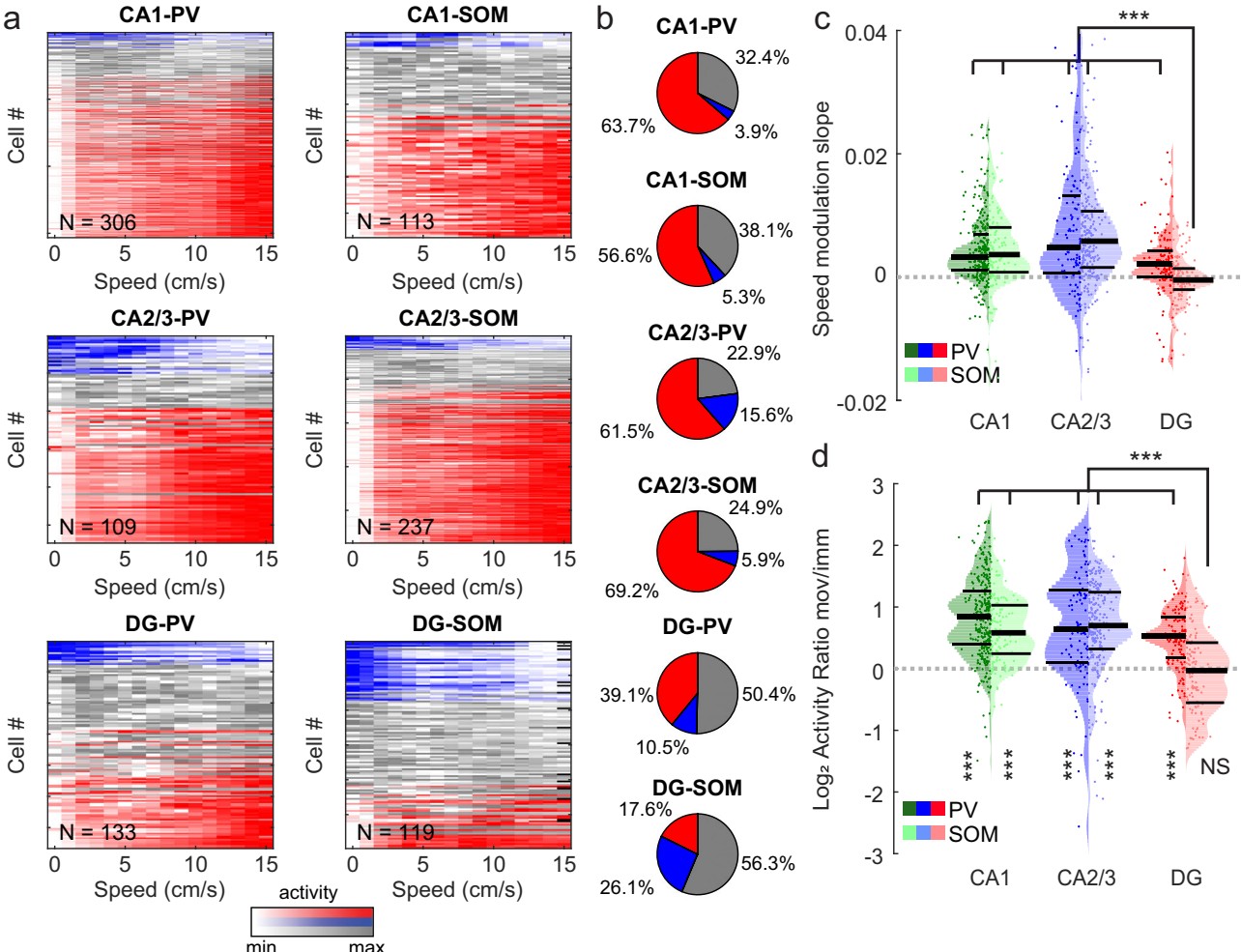

**Fig. 2 | Positive and negative speed modulation of interneuron activity.**
**a** Calcium activity (peak-normalized) over animal running speed for PV- and SOM-INs in CA1 (top), CA2/3 (middle) and DG (bottom). Color code denotes significance of speed modulation (red = positive, blue = negative, gray = no significant speed modulation) for individual cells. **b** Fraction of cells with significant positive (red), negative (blue) or no (gray) speed modulation for the individual IN types and hippocampal areas. **c** Distribution of slopes for speed modulation obtained by a linear fit to the data (see Fig. 1f, j blue dotted line) for PV-cells (dark colors) and SOM-cells (light colors) in the three hippocampal areas. **d** Ratio of activity-rates between phases of moving and immobility for PV-cells (dark colors) and SOM-cells (light colors) in the three hippocampal areas. Vertical markup denotes significance of the paired comparison of moving vs. immobile activity (Wilcoxon signed rank test). $p$ values between groups (horizontal) in **c**, **d** represent Kruskal–Wallis- with Dunn's post hoc test. Lines represent median and interquartile ranges, dots represent data from individual cells. NS not significant; ***$p < 0.001$. All statistical tests are two-sided. For exact $p$ values and comparisons between all groups see Supplementary Table 2.

SOM-INs in subfield-specific responses to novelty, we compared their activity during movement in familiar and novel environments (Fig. 4). Published reports on CA1 PV-IN responses to novelty are heterogeneous: Some studies reported reduced somatic activity[37,39], while others found increased activity of their axonal terminals in the CA1 pyramidal cell layer[38]. Although our data indicate a minor reduction in mean calcium activity of CA1 PV-INs in the novel environment, the effect was not statistically significant ($p = 0.38$; Fig. 4a, b, *dark green*). Consistent with previous observations in CA1 SOM-INs[37–39], we found a significant reduction of their mean activity when mice transitioned to the novel environment (Fig. 4a, b, *light green*). CA2/3 INs showed more distinct changes, characterized by marked activity reductions in both PV- and SOM-INs (Fig. 4a, b, *blue*). Similar to their counterparts in CA2/3, DG PV-INs drastically lowered their activity in the novel environment, indicating reduced perisomatic inhibition (Fig. 4a, b, *red*). Surprisingly, DG SOM-INs displayed opposite changes characterized by a marked activity increase and, thus, elevated dendritic inhibition in the novel environment (Fig. 4a, b, *pink*). These

observations in DG INs were largely consistent when data were aggregated over sessions or animals (Supplementary Fig. 2f). The activity increase of DG SOM-INs in novel environments was significantly different from the activity reduction of all other IN populations (Supplementary Fig. 3h).

DG SOM-INs comprise a heterogeneous set of subtypes[21]. Hilar perforant-path associated (HIPP) cells innervate the distal dendrites of GCs and INs in the molecular layer, thereby controlling input from the medial entorhinal cortex via the perforant path[13,30]. In contrast, local hilus-associated SOM-interneurons (HIL) form projections to distant brain areas, such as the medial septum, and their local axon collaterals are confined to the DG hilus[21]. To assess whether one, or both of these SOM-IN subtypes contributes to increased inhibition in novel environments, we exploited the differential distribution of their axonal arbors and selectively imaged the activity of SOM-IN axons in the hilus and the molecular layer, respectively, using an axon-targeted version of the GCaMP6f calcium indicator[59] (Fig. 4c, d; Supplementary Fig. 6a–f). Intriguingly, activity increases in the novel environment

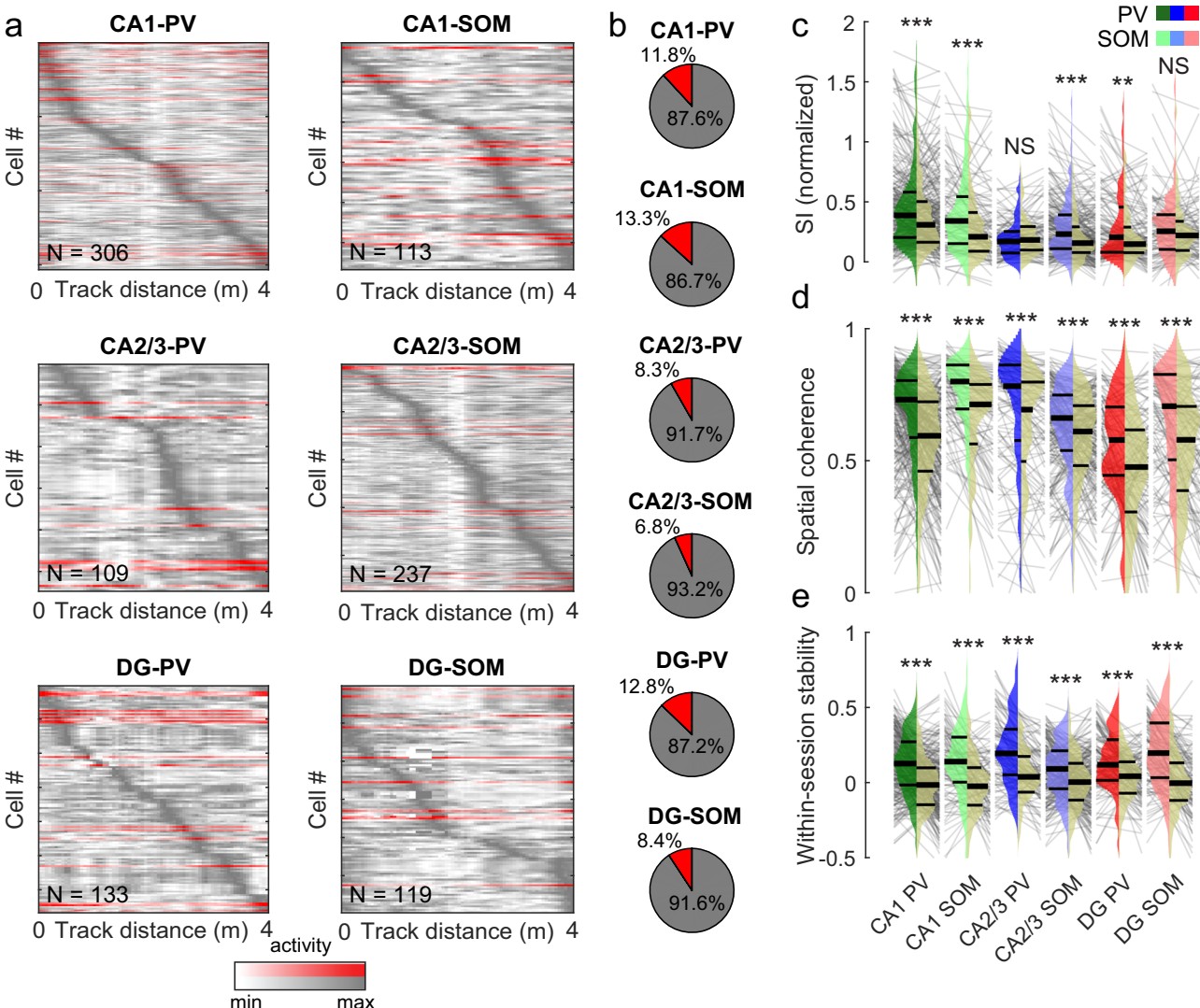

**Fig. 3 | Moderate spatial modulation of interneuron activity. a** Calcium activity (peak-normalized) over animal position for PV- and SOM-cells in CA1 (top), CA2/3 (middle) and DG (bottom). Red coloring denotes individual cells with significant spatial tuning. Cells were sorted by their maximum activity along the track. **b** Fraction of cells with significant spatial tuning (red) and un-tuned cells (gray). **c** Spatial information (SI) content normalized by the mean activity (left, colored) vs. bootstrap values (right, beige; see Methods) for PV- and SOM-cells in the three hippocampal areas. **d** Same as in c for spatial coherence (see Methods). **e** Same as in c, for stability of spatial tuning curves between the first- and second half of each session (see also Supplementary Fig. 5). **c–e** Wilcoxon signed rank-sum test. Thick lines represent median and interquartile ranges, thin lines in background connect data from individual cells and their respective shuffles. NS not significant; **$p < 0.01$, ***$p < 0.001$. All statistical tests are two-sided. For exact $p$ values and statistical comparison of values between IN groups see Supplementary Table 2.

were selective for the molecular-layer axons of putative HIPP cells (Fig. 4c; Supplementary Fig. 6a–c, g), while hilar axon collaterals of HIL cells showed reduced activation (Fig. 4d; Supplementary Fig. 6d–g), similar to the somatic activity of other hippocampal IN types (Fig. 4a, b). Thus, our data indicate a selective suppression of perforant path-mediated inputs by HIPP cell-dependent dendritic inhibition, which may limit the recruitment of other DG INs and GCs in novel environments[21,60].

Running speed profiles were highly similar between the familiar and the novel environment (Supplementary Fig. 3d, f, g) and mean running speeds were stable over the course of behavioral sessions (Supplementary Fig. 3e), making it unlikely that the novelty-associated differences were merely caused by a correlation of IN activity with the animals' mobility. Thus, our data suggest that inhibitory circuits may differentially gate information flow through the hippocampal formation in novel vs. familiar environments[6]. Particularly, differential novelty-induced modulation of somatic and dendritic inhibition in the DG could gate the GCs response to novelty[9,10].

## Novelty-dependent effects of chemogenetic IN suppression

To test our hypothesis that the novelty response of GCs is shaped by differential modulation of PV- and SOM-INs, we expressed modified human muscarinic M4 (inhibitory) receptors (hM4Di), tagged with the red fluorescent protein mCherry (AAV2.hSyn.DIO.hM4Di.mCherry), in the dorsal DG of PV-Cre and SOM-Cre mice (Fig. 5a–c). They enable reversible silencing of the transfected cells by injecting animals with the hM4Di agonist clozapine[61]. These animals were further injected with AAV1.Syn.GCaMP6f to express the calcium indicator pan-neuronally in the DG and image GC activity before and after chemogenetic inhibition of PV- and SOM-INs.

First, we examined the effects of clozapine application on the activity of DG PV- and SOM-INs identified by the mCherry tag and found a profound reduction of their respective activity levels after clozapine application (Fig. 5d). We then tested the effect of IN silencing on place cell activity in familiar and novel virtual environments (Fig. 6a, e). In line with the stronger activation of DG PV-INs in familiar environments, effects of PV-IN suppression were greater on the familiar

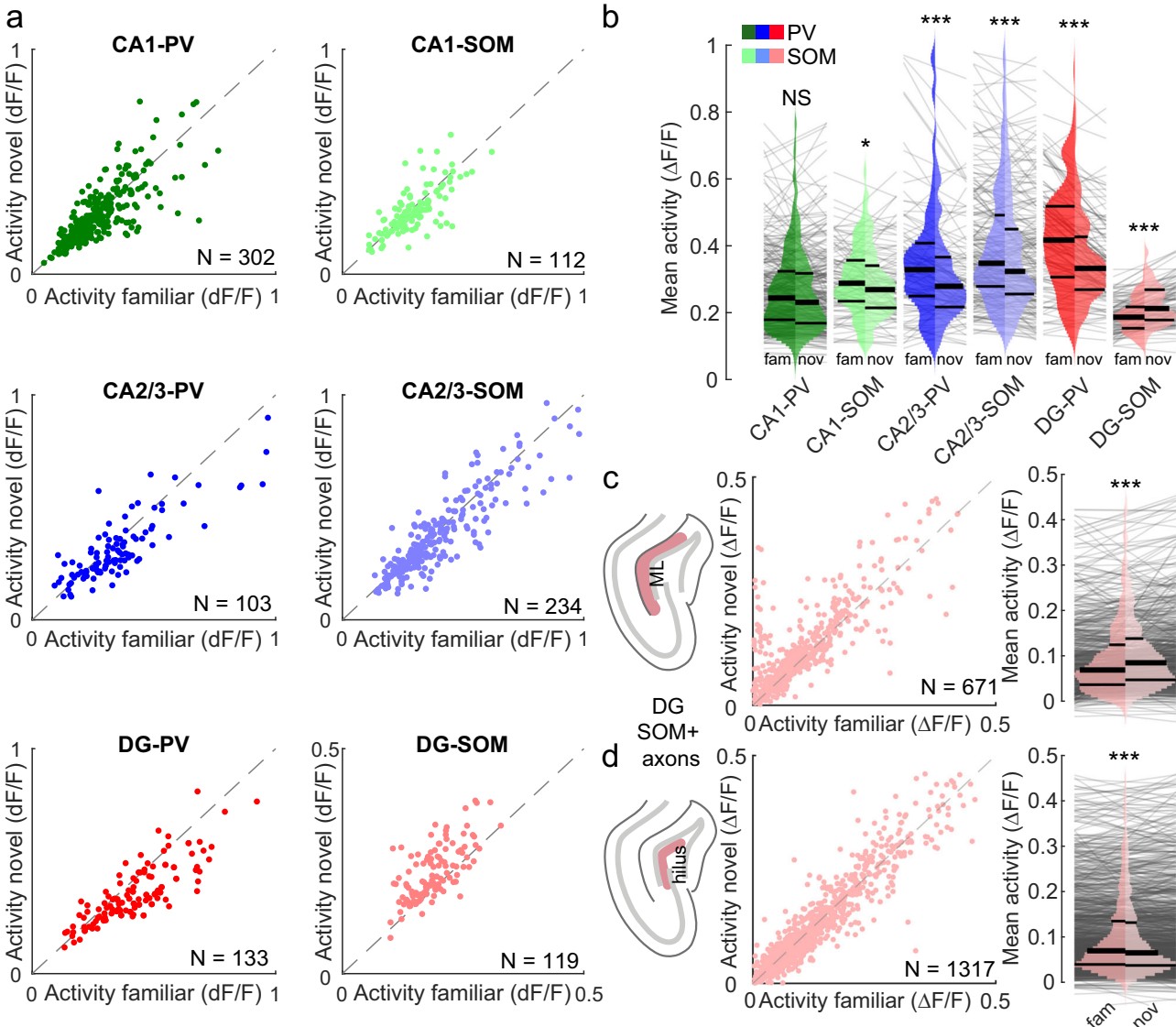

**Fig. 4 | Hippocampal area-specific modulation of interneuron activity in the novel context. a** Mean activity during movement in the familiar (*x-axis*) and novel (*y-axis*) context. Dots represent individual cells. **b** Distribution of mean activity rate during movement in the familiar (*left histograms, dark colors*) and the novel (*right histograms, light colors*) environment. **c** Same as in a (*middle*) and b (*right*) for DG SOM-IN axon terminals recorded in the DG molecular layer (ML; see schematic *left*).

**d** Same as in c, but for DG SOM-IN axon terminal recorded in the DG hilus. b,c,d Wilcoxon signed rank test. Thick lines represent median and interquartile ranges, thin lines in background represent data from individual cells or axons, respectively. NS, not significant; *$p < 0.05$; ***$p < 0.001$. All statistical tests are two-sided. For exact $p$ values see Supplementary Table 2.

than on the novel track (Fig. 6b–d). Intriguingly though, chemogenetic silencing of PV-INs led to a net reduction of GC activity in the familiar environment (Fig. 6c), indicating that their physiological effect on most GCs in familiar environments could be disinhibitory. This finding was consistently observed when data were aggregated per session ($p = 0.0043$), though not per animal ($p = 0.094$; Supplementary Fig. 2g). We further observed a significantly increased variance of GC activity rates in the familiar environment after blocking PV-mediated inhibition ($F = 2.2$, $p = 0.00012$, two-tailed F-test), which was neither present in the novel environment ($F = 1.35$, $p = 0.19$, two-tailed F-test), nor after disrupting DG SOM-IN activity ($F = 1.17$, $p = 0.25$, two-tailed F-test; Fig. 6c). This selective increase in GC activity-rate variance during conditions of suppressed PV-IN mediated inhibition may indicate a diversification in GC firing with a concomitant reduction of activity rates in most cells, but stagnant or increased rates in a smaller subpopulation. Unlike in the DG, chemogenetic suppression of CA1 PV-INs led to marked increases of pyramidal cell activity in the familiar and

novel environment (Supplementary Fig. 8a, b, k), in line with their lower modulation by behavioral novelty (Supplementary Fig. 3h). A 3-way ANOVA indicated a significant region- and genotype specific effect of PV-IN inactivation on principal cell transient rates ($p = 0.0429$; Supplementary Fig. 8k, Supplementary Table 2). Our results therefore indicate that DG PV-INs exert a familiarity-dependent disinhibitory effect on GCs while CA1 PV-INs exert a net-inhibitory effect on their surrounding principal cells.

Effects of pharmacogenetic DG SOM-IN inactivation on GC activity appeared to be different from those of PV-IN inactivation. Indeed, a 2-way ANOVA indicated a significant interaction between GC activity, IN type, and novelty ($p = 0.0425$; Fig. 6l; Supplementary Table 2). Unlike during DG PV-IN inactivation, we did not observe changes in GC activity rates in the familiar environment when DG SOM-INs were suppressed (Fig. 6g, l). However, we did observe a significant increase in GC activity rates in the novel environment after suppressing DG SOM-IN activity with clozapine (Fig. 6g *right*), although the size of the

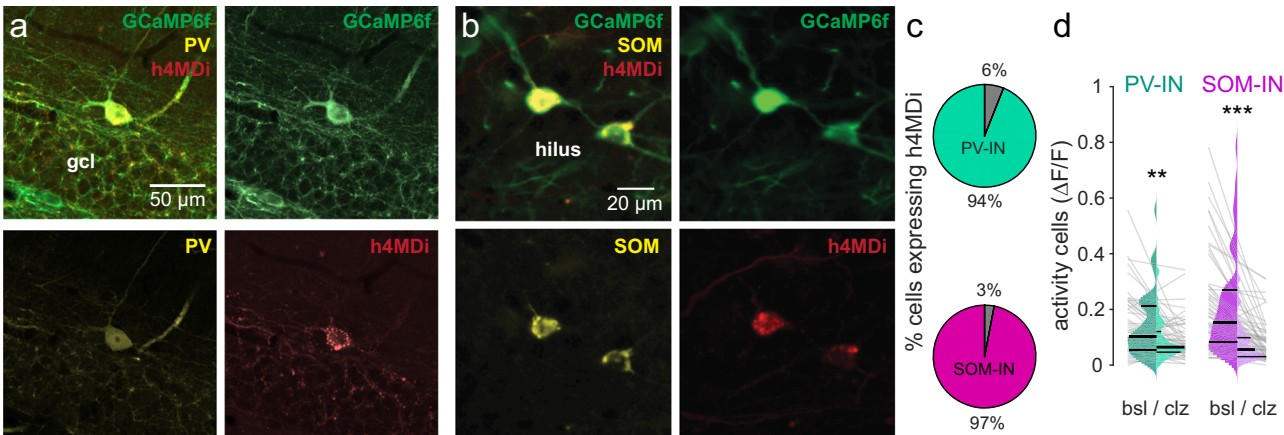

**Fig. 5 | DREADDs effectively silence dentate gyrus interneurons. a** Post-hoc confocal fluorescence image of a brain slice from a PV-Cre animal injected with floxed h4MDi (red) and non-floxed GCaMP6f (green; Methods) together with PV immunolabelling (yellow). Section is representative for $n = 6$ sections obtained from $N = 3$ animals. **b** Same as in a but for a SOM-Cre animals. Section is representative for $n = 6$ sections obtained from $N = 3$ animals. **c** Fraction of DG PV-INs (green) and DG SOM-INs (magenta) identified by immunolabelling that were co-expressing h4MDi after injection of the double-floxed virus in the respective Cre-animals (Methods). **d** Activity of PV- and SOM-INs before (left half) and after (right half) application of clozapine in PV- and SOM-Cre mice transfected with h4MDi, respectively. Lines indicate data from individual cells. All statistical tests are two-sided. For exact $p$ values see Supplementary Table 2.

change appeared to be small and was not significantly different from that observed in PV-Cre animals or control animals injected with an inactive viral construct in the same conditions (Fig. 6i–l). Chemogenetic suppression of SOM-IN activity in CA1 led to a significant increase of pyramidal cell transient rates in both familiar and novel environments (Supplementary Fig. 8f, g), although the size of this effect was not statistically different between brain regions (Supplementary Fig. 8k). The overall effect of DG SOM-INs on GC activity rates therefore seemed limited and was only significant in pair-wise statistical comparisons of activity rates in the novel environment.

In line with our previous observations[10], DG GC activity rates and SI were lower in the novel, compared to the familiar environment in both SOM-Cre and PV-Cre mice under control conditions (Supplementary Fig. 7a, b blue). We found that this difference vanished after chemogenetic silencing of either DG PV- or SOM-INs, albeit in different manners: While PV-INs seemed to promote greater activation of most GCs in the familiar environment (Fig. 6c, l, m), the effect of DG SOM-IN suppression on GC activation was tentatively greater in the novel environment (Fig. 6g, h, l, m).

DG PV-IN inactivation significantly reduced the total fraction of GC place cells in the familiar environment in a pair-wise comparison (Fig. 6b), although a 2-way ANOVA analysis did not indicate a significant interaction between the clozapine effect on the fraction of active cells and novelty (Supplementary Table 2). This was accompanied by a reduced SI in GCs in both environments (Fig. 6d, m), indicating that perisomatic inhibition supports shaping the spatial tuning of GCs. This effect was specific to DG PV-INs since chemogenetic suppression of CA1 PV-IN activity had the opposite effect on pyramidal cells (Supplementary Fig. 8c, l). Indeed, a 3-way ANOVA indicated a significant interaction between IN-type, brain region and spatial information (Supplementary Table 2). SI in the familiar environment was significantly elevated after CA1 PV-IN silencing (Supplementary Fig. 8c, l). Unlike PV-IN silencing, suppression of DG SOM-INs did not markedly alter the place-cell fraction among GCs (Fig. 6f) or their SI in either environment (Fig. 6h), while suppression of CA1 SOM-IN activity led to minor reductions in CA1 pyramidal cell SI (Supplementary Fig. 8h, l).

In a separate cohort of PV-Cre and SOM-Cre mice that expressed pan-neuronal GCaMP6f and an IN-type specific, inactive viral construct encoding mCherry, but not h4MDi, we observed no significant changes in the mean GC activity, SI, or place field correlations between familiar and novel contexts after clozapine injection (Fig. 6i–k). These data indicate that our identified effects of chemogenetic SOM and PV-IN silencing onto principal cell activity were due to h4MDi-driven silencing of the respective IN type rather than caused by unspecific clozapine effects. Taken together, our data suggest that DG PV-INs help to shape the spatial tuning of GCs and exert a moderate disinhibitory effect on GCs in familiar environments, where PV-INs are most active. This appears to be a DG-specific effect and opposite to that of CA1 PV-INs. In contrast, the effect of DG SOM-INs is modest and significant only in pair-wise statistics, but tentatively seems to limit GC activation during behavioral novelty where DG SOM-IN activity levels are increased.

## Silencing of DG SOM- and PV-INs differentially affect remapping

Given the differential impact of PV- and SOM-IN inhibition, we next asked whether IN inactivation affected the way individual GCs change ('remap') their spatial activity profiles, or 'place fields', between the two contexts[6]. While PV-IN silencing reduced GC activity in the familiar environment (Fig. 6c), it also reduced the level of overlap in spatial activity profiles between the familiar and novel environment (Fig. 7a). Thus, the correlation of GC place fields between contexts were reduced after DG PV-IN suppression (Fig. 7b). A population-vector analysis (Supplementary Fig. 7d) did not support this conclusion but was also significantly impacted by the fact that pairs of population vectors before and after clozapine could only be constructed for 5 out of 15 recording sessions due to the low numbers of simultaneously recorded active GCs. In contrast, chemogenetic silencing of CA1 PV-IN led to increased correlations of pyramidal cell place fields between contexts (Supplementary Fig. 8d, e).

In contrast to PV-IN, DG SOM-IN inactivation increased the correlation of GC place fields between the familiar and novel context (Fig. 7c, d) and led to significant increases in GC population vector correlations between contexts after clozapine application (Supplementary Fig. 7d). This is in line with published observations of increased contextual overlap of GC assemblies expressing immediate-early genes (IEGs) after silencing of DG SOM-INs[62]. Similarly, increased overlap of pyramidal cell place fields was observed in CA1 after inhibition of local SOM-INs (Supplementary Fig. 8i, j). Notably, GC place field correlations between the familiar and novel context remained

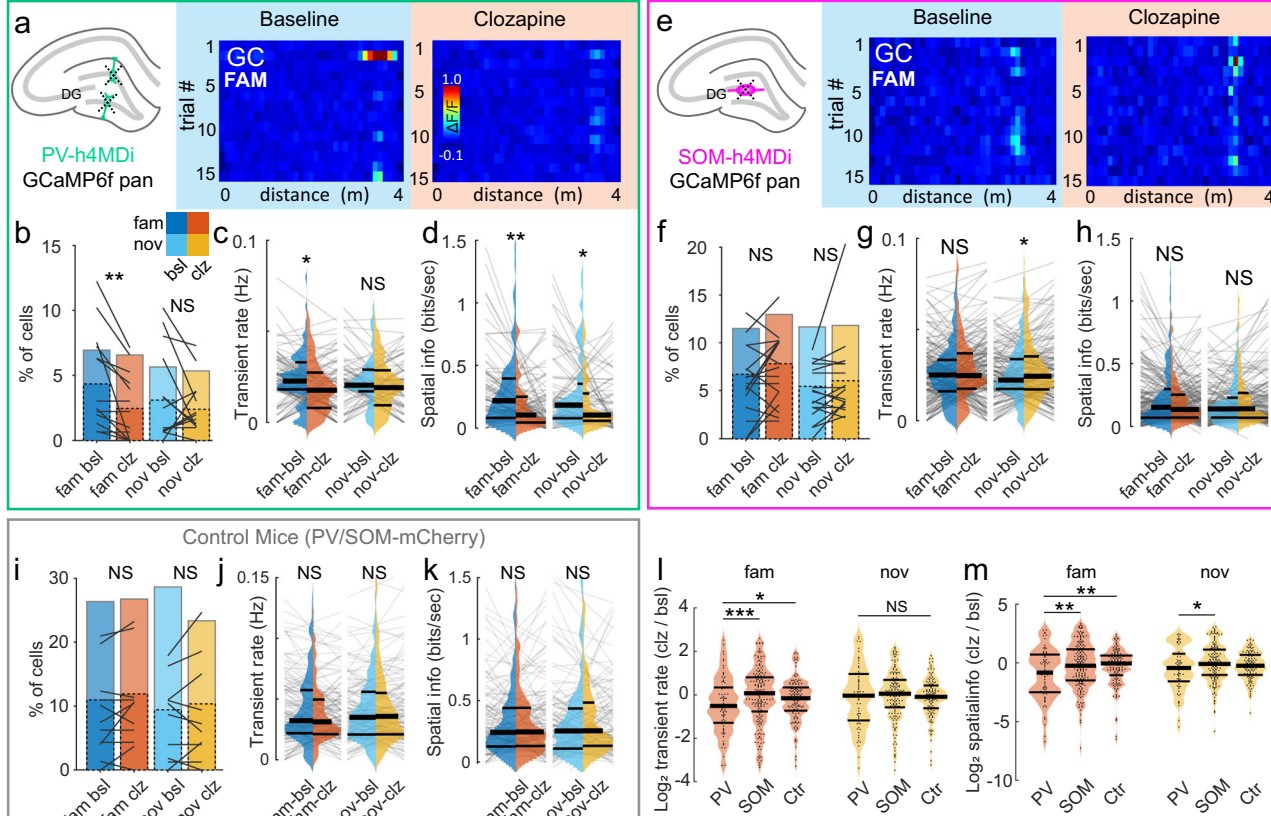

**Fig. 6 | Effects of interneuron silencing via h4MDi on dentate granule cells.**
**a** Schematic of the experiment (*left*) and example activity of a DG-GC recorded during multiple trials on the familiar track before (*middle*) and after (*right*) injection of clozapine in a mouse expressing inhibitory hM4Di receptors in DG PV-INs.
**b** Average fraction of active cells (*bars with continuous outline*) and fraction of place cells (*bars with dotted outline*) in control (*blue*) and clozapine (*orange*) conditions in PV-h4MDi mice. Lines denote place-cell fractions in individual experiments.
**c** Average calcium-transient rate of GCs with place fields on the familiar track (*left group*) or novel track (*right group*) in control (bsl) and clozapine (clz) conditions in PV-h4MDi mice. **d** Same as in c but for spatial information in PV-h4MDi mice.
**e–h** Same as in **a–d**, but for mice where hM4Di was expressed in DG SOM-INs. See Supplementary Fig. 8 for the same experiment in CA1. **i–k** Same as in **b–d**, but for control PV-Cre and SOM-Cre animals in which INs were infected with the fluorescent protein mCherry. Injection of clozapine in these animals did not induce changes in GC activity. **l** Ratios of place cell transient rates after clozapine

application divided by those under baseline conditions. *Left*, brown ensemble shows ratios for familiar context place cells in PV-Cre (*left violin*) and SOM-Cre (*middle violin*) animals injected with the active h4MDi construct. Ctr (*right violin*) shows data from control mice injected with the inactive viral construct. *Right*, yellow ensemble shows the same but for novel-context place cells. **m** Same as in l but showing ratios of spatial information after clozapine application divided by baseline values. **c**, **d**, **g**, **h**, **j–m** Thick lines represent median and interquartile ranges. Lines or dots, respectively, in the background indicate datapoints from individual GCs. **b**, **c**, **f**, **g**, **i**, **j** paired t-test, **d**, **e**, **k** Wilcoxon signed rank-sum test, **l**, **m** Kruskal–Wallis ANOVA on Ranks with Dunn's post-hoc test. See the corresponding section of Supplementary Table 2 for a 2-way ANOVA testing for interactions between IN type and novelty. NS not significant, *$p < 0.05$; **$p < 0.01$; ***$p < 0.001$. All statistical tests are two-sided. For exact $p$ values see Supplementary Table 2.

constant before and after clozapine application in control PV-Cre and SOM-Cre animals injected with the inactive mCherry construct (Fig. 7e, f).

In summary (Fig. 8), our data indicate that the output of most hippocampal INs promotes remapping and decorrelation of principal cell place fields between familiar and novel contexts. DG PV-INs appear to be an exception from this rule: Their disinhibitory effect in familiar environments seems to promote a stronger overlap of place fields or context-generalization[6,63].

## Discussion

In this study, we recorded and manipulated the activity of PV- and SOM-expressing INs in different hippocampal subfields in mice exploring familiar and novel virtual environments. While prior studies have reported basic firing characteristics of unidentified fast-firing INs in the DG[44,45,64] and novelty-responses of four unclassified hilar and GC-layer INs[9], we report here for the first time behavior-related activity characteristics and network impacts of a representative sample of genetically identified DG PV- and SOM-INs, and compare them to their

counterparts in other hippocampal subfields. However, given the heterogeneity we observed between individual examples from PV- and SOM-INs, future studies will be needed to investigate the precise roles of anatomical and functional subtypes within those broader PV- and SOM-IN families[39,41]. Functional heterogeneity within these families may further contribute to the relatively subtle effects we observed in chemogenetic manipulation experiments, despite the overall effective suppression of PV- or SOMI-IN activity (Fig. 5d).

We discovered several unique behavior-related idiosyncrasies of DG SOM-INs (Fig. 8): Hilar SOM-INs more commonly show negative speed modulation than other types of hippocampal INs (Fig. 2). Furthermore, the response of DG SOM-cells to novelty is inverse to that in CA1 and CA2/3 (Fig. 4a, b; Supplementary Fig. 3h) potentially suggesting that recruitment or synaptic plasticity in DG cells through entorhinal cortex inputs is suppressed by increased dendritic inhibition in novel environments (Fig. 8b)[24]. In line with this idea, a potential minor increase in GC activation in the novel environment was observed after chemogenetic inhibition of DG SOM-INs (Fig. 6g), although this was not confirmed by further statistical testing outside of

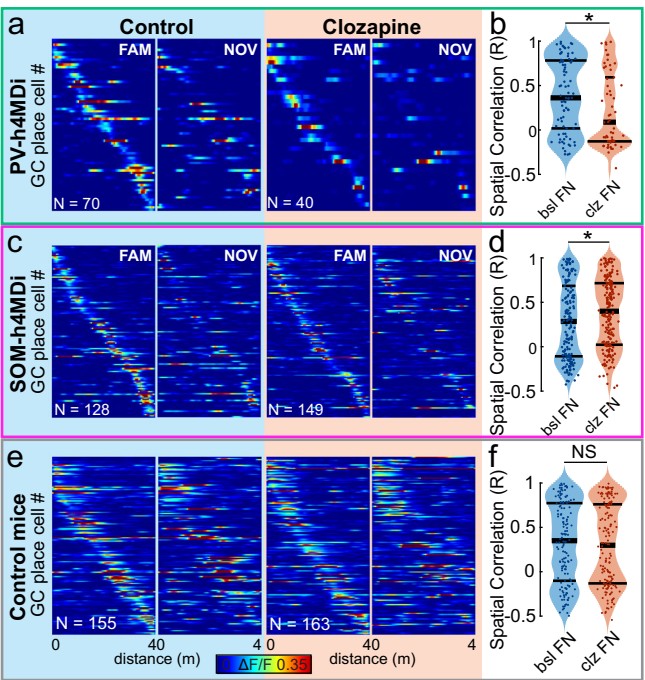

**Fig. 7 | Context-dependent remapping is differentially affected by SOM and PV interneurons. a** Activity maps of GC place cells are shown with the same sorting on the familiar (FAM; *left plot of the group*) and novel (NOV; *right plot*) track. Data are shown separately for cells with familiar context place fields during baseline conditions (bsl; *left group, blue shading*) and after clozapine injection (clz; *right group, orange shading*) in PV-Cre mice transfected with floxed h4MDi. **b** Correlations of spatial activity profiles ('place fields') between familiar and novel environments before (bsl F/N; *left, blue*) and after (clz F/N; *right, orange*) suppression of DG PV-IN activity via h4MDi. Note, decrease in place field correlations between familiar and novel environments after DG PV-INs silencing. **c, d** Same as in **a** and **b** but before (*left, blue*) and after (*right, orange*) disruption of DG SOM-IN activity through clozapine injection in SOM-Cre mice transfected with floxed hM4Di. Note, increase in place field correlations between familiar and novel environments after DG SOM-IN silencing. **e, f** Same as in **a** and **b** but for control PV- and SOM-Cre mice expressing an inert mCherry construct. Note, no changes in GC activity correlations between contexts were induced by clozapine alone. **b, d, f** Wilcoxon rank-sum test. NS not significant, *p < 0.05. All statistical tests are two-sided. For exact *p*-values see Supplementary Table 2.

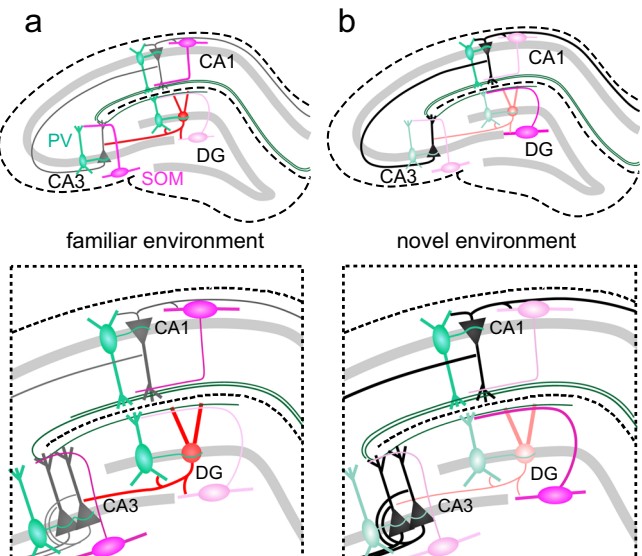

**Fig. 8 | Effects of hippocampal PV and SOM interneurons in familiar and novel environments. a** Schematic summary of interneuron activities and effects in the familiar environment. Green cells represent PV-INs, pink cells SOM-INs, red- and black cells depict DG GCs and hippocampal pyramidal cells, respectively. Green axon in the molecular layer illustrates perforant-path axons from the entorhinal cortex. Opacity of symbols indicates strength of activation. Relatively weak activation allows for stronger recruitment of DG GCs, while relatively stronger activation of other hippocampal neuron types keeps CA1-3 principal activation at bay. **b** Same as in **a**, but for novel environments. Increased activity of DG SOM-INs reduces the activation of other DG cell types, while relatively weaker activation of most other hippocampal IN classes may allow for stronger direct recruitment of pyramidal cells through the perforant path.

pair-wise comparisons (Fig. 6l). This came at the expense of increased overlaps of activity patterns between environments (Fig. 7c, d).

Finally, activity characteristics of DG PV-INs were remarkably similar to their counterparts in the *hippocampus proper*, but the effects of suppressing their output indicate that their impact on the surrounding principal cell network may be opposite to that of CA1 PV-INs (Supplementary Fig. 8k, l). Indeed, chemogenetic inactivation of DG PV-INs led to a paradoxical reduction in GC activity, accompanied by reduced spatial precision but more unique GC activity patterns for the respective environments (Figs. 6b–d and 7a, b). In contrast, suppression of CA1 PV-IN promoted increased pyramidal cell activation but a higher correlation of their place fields between environments (Supplementary Fig. 8a–e). Thus, our data indicate unique roles for both DG PV- and SOM-INs, which depend on behavioral novelty and are distinct from their respective counterparts in other hippocampal subfields.

In vivo electrophysiological studies revealed that discharges of CA1 PV- and SOM-INs are phase-coupled to theta and gamma network oscillations, which increase in power during running and foraging[65,66]. Synaptic inhibition is thought to be critical for the orchestration of hippocampal principal cells during fast network oscillations and to create temporal windows for synaptic plasticity to support the encoding of novel information[34,65,67,68]. Consistent with this proposal,

we observed a marked recruitment of most PV- and SOM-INs in CA1-3 during spatial exploration and a positive modulation of their activity by running speed (Fig. 2). However, we also detected a small proportion of negatively speed-modulated CA1-3 PV- and SOM-INs. These data fit to in vivo recordings of morphologically identified CA1 PV-expressing basket, axo-axonic and bistratified cells[69], which indicate within-type heterogeneities in the magnitude and sign of running-speed modulation[35,39,54,70]. In contrast to CA1-3 pyramidal cells, DG GCs are very sparsely active during spatial exploration but more of them become active during resting states like non-REM sleep[45,71]. Similarly, a substantial fraction of DG SOM-INs showed elevated activity during rest and declining activity with increasing running speed (Fig. 1j). This is particularly interesting because some hilar SOM-INs form long-range projections to distant brain areas, for instance the medial septum[21,72]. They could therefore be part of a broader network, which may also encompass other, less frequent IN types such as cholecystokinin-expressing basket cells[40], and support processing of internally generated information like thoughts or dreams, rather than external sensory perceptions.

Theoretical studies suggested that spatially tuned inhibition may shape spatial representations of temporal lobe principal cells[73], potentially inspired by sensory tuning of INs in other cortical areas[74]. However, subsequent experimental studies found that the inhibition received by principal cells is mostly uniform across space[34,75,76], although one recent study indicates inverse spatial tuning between synaptically connected interneuron-pyramidal cell pairs in CA1 that could be permissive for place field expression[77]. Our results replicate prior reports of low spatial tuning in CA1 PV- and SOM-IN[39], and indicate that spatial tuning of INs may be low across all hippocampal subfields (Fig. 3b), although the degree of spatial tuning on fine scales may be under-estimated in our experiments due to technical limitations of calcium imaging. However, even spatially uniform inhibition

can sharpen principal cell place fields by suppressing weak out-of-field excitation[76]. Particularly in similar environments, represented by overlapping afferent inputs, location independent inhibition may promote 'winner-takes-all' mechanisms[78]. A less excited principal cell ensemble encoding one representation can be efficiently suppressed by INs driven by a competing ensemble that receives slightly higher excitation and thereby ensures that only one representation is active at each time point[79]. In line with this idea, inhibition by CA1 SOM-INs, most of which mediate feed-back inhibition between CA1 pyramidal cells[13], appears to enhance CA1 spatial precision and their suppression lowers pyramidal cell SI (Supplementary Fig. 8h). Similarly, both CA1 SOM- and PV-INs appear to inhibit pyramidal cell activation related to other environments and therefore their chemogenetic inactivation increased contextual overlap of pyramidal cell place fields (Supplementary Fig. 8d, i). Strikingly, our data indicate a different division of labor in the DG, in which PV-INs may primarily control spatial precision of GCs (Fig. 6), whereas SOM-INs promote the establishment of more unique ensembles for different behavioral contexts (Fig. 7).

Although in vitro studies described a net inhibitory effect of DG PV-INs[80,81], in vivo studies found no reduced IEG expression in GCs after PV-IN silencing[62]. Our data indicate reductions in GC activity, spatial tuning, and place cell fractions after DG PV-IN silencing in familiar environments (Fig. 6b–d). This apparent excitatory effect of PV-IN output was unique to the DG and not observed in CA1 PV-IN (Supplementary Fig. 8b, k). It could be explained by depolarizing inhibition, which has been observed at PV-IN-to-GC synapses in vitro[82,83] and is attributed to extremely negative GC membrane potentials in vivo. Therefore, in contrast to brain-slice studies[81], the network effect of DG PV-IN-mediated GABA release could be excitatory, similar to previous reports of excitatory effects of cortical axo-axonic inhibition[84]. However, more recent in vivo studies of axo-axonic inhibition in CA1 indicate that this type of excitation may not be ubiquitously present or may be constrained to in vitro conditions[85], making this explanation less likely.

Another possibility is that PV-IN silencing enhances competitive, 'winner-take-all' dynamics within the GC population[78] by disinhibiting a small subgroup of GCs, which in turn recruit stronger inhibition through other INs onto the remaining GCs[30,62]. Similar complex network dynamics have been described for chemogenetic manipulations of CA1 IN activity[86]. This idea is further supported by the increased variance in GC activity rates after blockade of PV-, but not SOM-INs, which indicates an equalizing effect of PV-IN-mediated inhibition on activity rates across the GC population. GC-to-PV-IN connections show high recurrency and associative plasticity[26], which may gradually establish such equalizing dynamics over time[6] and explain stronger effects of PV-IN disruption in familiar environments. Finally, suppression of PV-INs may result in disinhibition of other DG IN types[30], which could suppress GC activity in the familiar environment below baseline levels. In summary, our findings indicate that PV-mediated inhibition in the DG has a complex impact on the GC network that appears to be disinhibitory in familiar environments.

Larger hippocampal activation by novel content has been reported in humans[87] and rodents[9], in line with its role in storing new memories. However, principal cells in individual hippocampal subfields respond differently to novelty[10–12,88]. Especially CA3 is critical for the rapid formation of novel contextual memories[11,88–90]. Reduced activity of PV- and SOM- INs in mice exploring novel environments suggests diminished perisomatic and dendritic inhibition in CA2/3 and CA1 (Fig. 8b), which in turn could support the formation of place fields representing the new environment by permitting active dendritic events in pyramidal cells to induce synaptic plasticity[38,77,91,92]. However, additional mechanisms, such as hippocampal area- and cell type-specific responses to neuromodulators like acetylcholine and dopamine[93,94] also likely play a role in promoting the encoding of novel afferent information[8,94,95]. Therefore, a novelty-induced reduction of

perisomatic and dendritic inhibition in CA1-3 (Fig. 4a, b), together with neuromodulatory mechanisms may promote the formation of novel place fields and associative encoding of memories in the hippocampus.

While PV-IN activity in novel environments was reduced or stagnant throughout the hippocampus, SOM-IN activity was reduced in CA2/3 and CA1, but markedly increased in the DG. This novelty-dependent increase in dendritic inhibition appears to be specific for axon terminals of putative HIPP SOM-INs in the molecular layer[22], where entorhinal cortex inputs from the perforant path contact GCs, but was absent in the hilar terminals of putative HIL SOM-INs[21] (Fig. 4c, d). While this effect appears overall subtle, caution is warranted with respect to the effect size as noise inherent to imaging subcellular structures deep in the brain will likely increase sample variance and could therefore significantly under-estimate the effect in the biological sample. Potent suppression of entorhinal inputs[60] by increased HIPP cell output may explain the reduced activity of GCs during exploration of novel environments[10] (Supplementary Fig. 7a). This increased dendritic inhibition may furthermore limit synaptic plasticity at entorhinal cortex inputs[8,96] in new environments and thereby protect highly stable GC place fields gradually formed through repeated exposure[10] from being 'overwritten'[97,98] (see also ref. [96]). Whether adult-born GCs at ~4-7 postnatal weeks of maturation, which receive the least lateral inhibition compared to older GCs[99] and show enhanced activity in response to novelty[100], may be particularly involved in the initial representation of novel environments, will require further investigations in future.

Finally, in contrast to DG PV-INs, SOM-INs appear to limit the number of GCs that are recruited into a neuronal ensemble in a novel environment and preserve sparse coding[62]. In line with this hypothesis, optogenetic silencing of DG SOM-, but not PV-INs increases the expression of the IEG *cfos* upon exploration of a novel environment and diminishes the animals' ability to discriminate similar contexts[62]. Similarly, chemogenetic suppression of SOM-IN activity in our experiments led to a higher overlap of GC place fields between environments (Fig. 7c, d). In summary, our data indicate that subregion-specific dynamics in hippocampal IN networks enable different processing modes for familiar and novel information (Fig. 8a, b), which are tailored to the specific principal cell representations in the respective hippocampal subfields[6,101]. Future studies are needed to determine whether these differences are due to intrinsic differences in morphology and function of PV- and SOM-INs in the respective areas[13] or inherited from external synaptic pathways or neuromodulatory systems.

## Methods

### Mice

All experiments involving animals were carried out according to national and institutional guidelines and approved by the 'Tierversuchskommission' of the Regierungspräsidium Freiburg (license no. G16/037) in accordance with national legislation. We used male PV-Cre (B6;129P2-Pvalbtm1(Cre)Arbr/J; The Jackson laboratory[102]) and SOM-IRES-Cre (Ssttm2.1(Cre)Zjh/J; The Jackson Laboratory[103]) mice aged 9–12 weeks at the beginning of experiments. Where possible, animals were recorded in more than one hippocampal region (see Supplementary Table 1), allowing us to reduce the number of animals for the IN imaging experiments. In those cases, imaging usually commenced in the more superficial region. In addition to the above, data from two B6;129P2-Pvalbtm1(cre)Arbr/J mice (PV-Cre; The Jackson laboratory) crossed with B6.Cg-Gt(ROSA)26Sortm9(CAG-tdTomato)Hze/J mice (Ai9-reporter; The Jackson laboratory) were included for the comparison with principal cell activity in (Supplementary Figs. 4 and 5) which have been previously published and are described in ref. [10]. All mice were housed on a 12-h light-dark cycle in groups of 2-3 mice at an ambient temperature of 20–24 °C and

45–65% humidity. No statistical methods were used to predetermine sample size. The experiments were not randomized, and the investigators were not blind to allocation during experiments and outcome assessment.

## Virus injections and head plate implantation

All surgical procedures were performed in a stereotactic apparatus (Kopf instruments) under anesthesia with 1·2% Isoflurane and analgesia using 0.1 mg kg$^{-1}$ buprenorphine. A small (~0.5–1 mm diameter) craniotomy was made over the hippocampus and 500 nl of different combinations of AAVs were injected into CA2/3 (A/P: −1.7 mm; M/L 1.9 mm; D/V −1.9 mm) or DG and CA1 (A/P: −2.0 mm; M/L 2.0 mm; D/V −2.0 and/or −1.4 mm) depending on the intended experiment. For imaging of interneuron activity, AAV1-CAG-FLEX-mRuby2-GSG-P2A-GCaMP6f-WPRE-pA (titer 4*10$^{13}$ vg ml$^{-1}$; University of Pennsylvania Vector Core) was injected either in SOM-Cre or PV-Cre mice. For imaging of DG SOM-IN axonal activity AAV9.hSyn.flex.GAP43-GCaMP6s (titer 1*10$^{12}$ vg ml$^{-1}$; University of Pennsylvania Vector Core) was injected into the DG of SOM-Cre mice. For chemogenetic manipulation experiments with simultaneous imaging of principal cell population activity, AAV2-hSyn-DIO-hM4D(Gi)-mCherry (titer 2.7*10$^{12}$ vg ml$^{-1}$: University of Pennsylvania Vector Core) was injected either into the DG (Figs. 5–7; Supplementary Fig. 7) or CA1 (Supplementary Fig. 8), together with AAV1-Syn-Gcamp6f (titer 1.7*10$^{12}$ vg ml$^{-1}$: University of Pennsylvania Vector Core). For clozapine control experiments AAV2-hSyn-DIO-mCherry (titer 2.3*10$^{12}$; University of Pennsylvania Vector Core) and AAV1-Syn-Gcamp6f (titer 1.7*10$^{12}$ vg ml$^{-1}$: University of Pennsylvania Vector Core) was injected into the DG of 3 PV-Cre and 3 SOM-Cre mice. Together with the viral injection, mice were implanted with a stainless-steel head plate (25 × 10 × 0.8 mm with an 8 mm central aperture) in the same surgery session. The head plate was oriented horizontal for CA1/DG imaging implantations and with a 20° lateral angle for CA2/3 imaging implantations. Mice recovered from surgery for at least 5 days before training sessions commenced. Postoperative analgesic treatment was continued with carprofen (5 mg kg$^{-1}$ body weight) for 3 days after surgery.

## Imaging window implantation

Cortical excavation and imaging window implantation were performed >10 days after the initial virus injection, according to published protocols[10,48]. A craniotomy (diameter 3 mm) was made centered at A/P −1.5 mm and M/L −1.5 mm for CA1/DG imaging and A/P −1.5 mm, M/L −2.5 mm for CA2/3 recordings. For implantations over CA2/3, the head of the mouse was tilted by 20°, so that the implantation plane was parallel to the lateral part of the pyramidal cell layer. Parts of the somatosensory cortex as well as posterior parietal association cortex were gently aspirated while irrigating with chilled saline. We continued aspiration until the external capsule was exposed. The outer part of the external capsule was then gently peeled away using fine forceps leaving the inner capsule and the hippocampus itself undamaged. The imaging window implant consisted of a 3 mm diameter coverslip (CS-3R, Warner Instruments) glued to the bottom of a stainless-steel cannula (3 mm diameter 1.2–1.5 mm height). The window was gradually lowered into the craniotomy using a forceps until the glass was in contact with the external capsule. The implant was then affixed to the skull using cyanoacrylate. Mice were allowed to recover from window implantation for 2-3 days.

## Virtual environment setup and behavioral training

Our custom virtual environment setup, described in detail in[10], consisted of an air-supported polystyrene ball (20 cm diameter). A small metal axle was attached to the ball's side to constrain ball motion to the forward/backward direction. Motion of the ball was monitored with an optical sensor (G-500, Logitech) and translated into forward motion through the virtual environment. In later experiments,

specifically those for DREADD manipulations, the ball was replaced by an air-supported polystyrene wheel (18 cm diameter) to serve the same purpose. The forward gain was adjusted, so that 4 m of distance traveled along the circumference of the treadmill equaled one full traversal along the circular, or linear track, respectively. The virtual environment was displayed on four TFT monitors (19" screen diagonal, Dell) arranged in a hexagonal arc around the mouse and placed ~25 cm away from the head, thereby covering ~260° of the horizontal and ~60° of the vertical visual field of the mouse. The virtual environment was created and simulated using the open-source 3D rendering software Blender. The track consisted of textured walls, floors and other 3D rendered objects at the track's sides as visual cues. To motivate consistent behavior, soy-milk rewards (4 μl) were administered when the animal traversed certain locations on the track. For the familiar track, reward locations were fixed to two locations on every lap while for the novel tracks, rewards were dispensed in two out of four potential locations and only every other location was rewarded. The first rewarded location on the novel track was chosen at random but kept consistent throughout a given imaging session encouraging alternating behavior. Visual cues on the track signified potential reward locations to the animal and rewards were delivered as soon as the animal reached the rewarded locations. The amount of reward per lap was equal between the familiar and novel environment. Reward frequency was the same between training and test sessions. To make sure that all technical conditions of the imaging were identical between the familiar- and novel context runs, we alternated between the familiar and novel context during continuous imaging runs. In these, the animal ran first for 60 s on one track and was then 'teleported' from its current position to the start of the other track and able to run for another 60 s after a brief (<10 s) pause, in which the screens were blanked. During this re-set, the system did not capture any behavioral information (including the animal's speed). Due to the display process, animals had an indication that a new trial was about to start 0.5 to 1 s before the trial started and behavioral information was captured.

Five days after head plate implantation, mice were placed in the virtual environment for 10–30 min daily, with gradually increasing timespans. During this time, only the familiar context was available to the mice. After 4-5 days of habituation, mice showed consistent running (Supplementary Fig. 3). Mice were thereafter implanted with the cortical window and allowed to recover from the surgery. After recovery, training in the virtual environment was re-initiated for 30–60 min daily in the familiar context until consistent running was observed in all animals and familiarization to the context had been achieved for at least 10 days in total before start of the imaging sessions.

From the first day of the imaging session, mice were introduced to a novel context, which had different visual cues, floor- and wall textures, but had the same dimensions as the familiar environment. There were several different novel contexts previously unseen by the animal and one of them was randomly selected for each experiment. Mice alternatingly ran on the two tracks for a total of 10 min on each track. In many of the mice, the visible area under the imaging window was sufficiently large to select another imaging field of view that was not overlapping with the first and contained a different population of neurons. In these cases, we repeated the entire imaging experiment on another day and used the new field of view and a different novel context (Supplementary Table 1).

## Chemogenetic suppression of IN activity

Imaging of GC place-coding before and after chemogenetic manipulation of IN activity in animals injected with the h4MDi construct was performed in line with our prior experiments[10]. Animals were first trained for several days on a linear track in the virtual environment setup and thoroughly familiarized with one of the tracks. On the first day of recordings, they were placed in the apparatus and GC activity

was imaged during a session consisting of 15 runs on the familiar track, followed by 15 runs on a novel track. In contrast to the procedure outlined above, sessions lasted until the animal had completed a fixed number of laps (rather than for a specified time-period) to allow for accurate assessment of spatial coding in the sparsely active GC population[10]. Immediately after this baseline imaging session, animals were removed from the behavior setup and injected intraperitoneally with clozapine (1 mg/kg; Tocris) and placed back in their home-cage for 30 min. They were then placed into the apparatus again and completed a second set of 15 laps on the familiar and the same novel track. In animals in which more than one high-quality field of view with GCs was accessible, the entire experiment was repeated on another day with the same familiar, but a different novel track (Supplementary Table 1).

### In vivo two-photon calcium imaging

Imaging was performed using a resonant/galvo high-speed laser scanning two-photon microscope (Neurolabware) with a frame rate of 30 Hz for bidirectional scanning and a power of 5–20 mW measured at the objective front-lens for all hippocampal subfields. The microscope was equipped with an electrically tunable, fast z-focusing lens (opto-tune, Edmund optics) to switch between z-planes within less than a millisecond. Images were acquired through a 16× objective (Nikon, 0.8 N.A., 3 mm WD), which was tilted at an angle of 20° for CA2/3 imaging. GCaMP6f was excited at 930 nm with a femtosecond-pulsed two-photon laser (Mai Tai DeepSee®, Spectra-Physics). We scanned three imaging planes (-25 μm z-spacing between planes) in rapid alternation so that each plane was sampled at 10 Hz. The planes spanned 300–500 μm in x/y-direction and were placed so that as many labeled neurons as possible were depicted. To block ambient light from the photodetectors, the animals head plate was attached to the bottom of an opaque imaging chamber before each experiment and the chamber was fixed in the behavioral apparatus together with the animal. A ring of black foam rubber between the imaging chamber and the microscope objective blocked any remaining stray light.

### Histology and detection of imaging area

At the end of experiments, mice were deeply anaesthetized using ketamine/xylazine (Sigma Aldrich) and image stacks of the area underneath the imaging window were acquired in vivo in the two-photon microscope. Mice were then perfused transcardially with 4% paraformaldehyde in phosphate-buffered saline (PBS). Brains were cut in 100 μm coronal slices and sections containing the area underneath the imaging window were collected. Image stacks of GCaMP6f and mRuby2 fluorescence in the sections were acquired with a confocal microscope (LSM 710, Zeiss) and the imaged region was re-identified by comparing these stacks with the ones obtained in vivo, as described[10]. In a subset of animals, immunohistochemistry was performed and slices were stained against PV (polyclonal rabbit, 1:1000, Swant; catalog no.: PV27, LOT 2014; validation in PV knockout mice), SOM (polyclonal rabbit, 1:500, Peninsula Laboratories; catalog no.: T4102, LOT A18PO21141; validation with Enzyme Linked Immunosorbent Assay (ELISA) and immunohistochemistry) and then counterstained with secondary antibodies goat-anti-rabbit coupled to AlexaFluor 647 (1:1000, Abcam; catalog no.: AB150079, LOT GR3444080-3; validated by immunohistochemistry). Antibodies were applied in PBS (3 slices in 0.5 ml) containing 0.3% Triton X-100, 3% NGS for 24 h at 4 °C and the secondary antibody for 3 h at 22 °C. For quantification of h4MDi expression (Fig. 6a, b), colocalization of SOM- or PV- immunoreactivity with mCherry labeling the h4MDi construct was assessed.

### Imaging data processing, segmentation and data extraction

Motion correction of all imaging data was performed line-by-line using the SIMA software package[104] with a 2D Hidden Markov Model or with the software-package 'Suite2P'[105]. If no decent motion correction could be achieved, the data were discarded. The motion-corrected and time-averaged image of mRuby2 for each run was used to align recordings from the same field of view relative to each other and their displacements were stored with the dataset.

To segment interneuron somata, regions of interest (ROIs) were drawn manually using ImageJ (NIH) or automatically by applying the 'suite2P' software package[105]. In case of automated ROI settings, individual ROIs were subsequently inspected by the experimenter. Cell bodies were identified based on the motion-corrected, time-averaged GCaMP6f and mRuby2 fluorescence images and re-inspected for each run to make sure that segmented cells were clearly visible throughout the experiment. The obtained ROIs were transformed according to the displacement between the mean mRuby2 fluorescence images as described above and the average calcium signal over time was obtained from each ROI for all runs. We restricted our analysis to running periods of mice with a minimum speed of 2 cm*s$^{-1}$. We chose this threshold, rather than strictly zero, for immobility because there is a minor jitter of the position signal in our recordings even when the animal is sitting still on the ball that is caused by motion of the animal's head (e.g. during grooming, whisking, stretching, etc.) relative to it's paws on the ball but not related to locomotion or directed movement.

For hippocampal principal cells, individual transients were detected as previously described[10,48]. In brief, calcium traces were corrected for slow drift by subtracting the 8th percentile value of the fluorescence-value distribution in a window of -8 s around each time point from the raw fluorescence trace. We obtained an initial estimate on baseline fluorescence and standard deviation (SD) by calculating the mean of all points of the fluorescence signal which did not exceed 3 standard deviations (SD) of the total signal and would therefore likely belong to transients. We divided the raw fluorescence trace by this value in order to obtain a ΔF/F trace. We used this trace to determine the parameters for transient detection that yielded a false positive rate (defined as the ratio of negative to positive going transients) <5% and extracted all significant transients from the raw ΔF/F trace. Definitive values for baseline fluorescence and baseline SD were then calculated from all points of this trace which did not contain significant transients. For further analysis, all values of this ΔF/F trace that did not contain significant calcium transients were masked and set to zero. Principal cell transient rates were computed for periods of movement with speeds >2 cm/s.

For hippocampal INs, rigorous identification of spike or burst-related calcium transients is often prevented by their high firing rates, which exceeds the acquisition rate of our imaging system and the kinetics of the calcium indicator[44,47,106]. However, the calcium signal from INs can still be used to approximate their firing rate over time[36,38,39]. To obtain baseline-normalized ΔF/F calcium traces, we examined the fluorescence value distribution from each recording segment (containing 60 s familiar-track and 60 s novel-track running and a brief blank interval) and divided the entire trace by the 8th percentile value of this distribution[38].

### Speed modulation

To determine speed modulation of individual cells, mean GCaMP6f fluorescence (ΔF/F) was determined as a function of running speed in bins of 1 cm*s$^{-1}$ (Fig. 1e, i). These values were fitted with a linear function and the slope was used as a measure for the sign and magnitude of speed modulation. Speed tuning was considered significant for a cell if there was a significant correlation (Pearson's $R$) between fluorescence signal and running speed.

### Spatial tuning parameters

For calculating spatial information (SI), the average calcium activity (mean ΔF/F) was computed for each 5 cm wide bin along the linear track and used as an approximation for the neurons average firing rate in that location. SI was then calculated for each cell as SI =

$\sum_{i=1}^{N} \lambda_i \log_2 \frac{\lambda_i}{\lambda} \, p_i$ where $\lambda_i$ and $p_i$ are the average calcium activity and fraction of time spent in the $i$-th bin, respectively, $\lambda$ is the overall calcium activity averaged over the entire linear track and $N$ is the number of bins on the track. For INs and principal cell to IN comparisons on Supplementary Figs. 4 and 5, we further divided this value by the mean activity of the respective neurons (as the mean integral, or area under the curve (AUC), of the fluorescence signal over time) to approximate the spatial information per unit activity (bits $* \Delta F/F^{-1}$), conceptually similar to the conventional bits $*$ spike$^{-1}$ metric used in electrophysiological recordings. To compute spatial vector tuning (Supplementary Fig. 4), we plotted the mean activity ($\Delta F/F$) of each spatial bin at its respective angle from the start position on the circular track into a polar coordinate system (Supplementary Fig. 4a). We then computed the circular mean of this distribution to obtain the cell's mean tuning vector length and angle. Spatial coherence was determined as the correlation (Pearson's $R$) between the mean fluorescence value in each 5 cm bin on the track with its two nearest neighbors to give a measure for the local smoothness of the spatial tuning curve[56].

### Within session stability of spatial representations

To assess the stability of a place cell's spatial representation within a session, we divided the track into 5 cm bins and calculated the mean $\Delta F/F$ value for each bin while the animal was moving on the track with a speed >2 cm/s to get activity maps for each individual cell. This was done separately for the first- and second half of the recording session. We then computed the within-session stability as the cross-correlation between the mean activity maps of the first and second half of the session (Fig. 3E; Supplementary Fig. 5). We also computed population vector correlations as a function of position in the first and second half of the recording (Supplementary Fig 5b, e) to visualize the local similarity of population activity across time. Before computing these correlations, we re-normalized each neuron's map by subtracting the mean over space and dividing by the standard deviation (z-scoring) to mitigate potential effects of mean rate differences between cells on the assessment of local population vector similarity.

### Bootstrap analysis of spatial tuning parameters and place-cell detection

To determine whether spatial tuning was significant for individual cells (Fig. 3a) or cell groups (Fig. 3c–e), we generated bootstrap values for each cell by circularly shuffling the animal's position on the track relative to the calcium activity trace of this cell in 1000 random intervals. For this procedure, the position trace and the calcium trace, respectively, were concatenated across all runs under a given condition and then the position trace was shifted forward by a defined interval. The missing values at the beginning of the position trace, introduced by the shift, were then replaced with those 'falling off' at the end of the position trace resulting in a circular shift of position data. For each of these random shuffles, SI, spatial coherence, tuning-vector length, and within-session stability were re-calculated. Tuning of a given parameter was considered significant if a cell's true value exceeded the random distribution in more than 95% of shuffles. To analyze whether tuning was significant across the population, we randomly selected one of these bootstrapped values for each cell and compared the obtained distribution of random values with the respective true values with paired statistical testing. For the analysis of principal cell activity in our chemogenetic IN manipulation experiments (Figs. 6, 7, and Supplementary Figs. 7, 8), we defined a cell as place-cell if it had significant SI as defined above and a minimum activity rate of 1 transient per minute. In line with previous publications[10,107], the vast majority of GCs was silent during each recording. To limit our comparisons to the relevant cells, we selected populations of GCs for all group comparisons that had a place field in at least one of the compared conditions.

### Between context remapping of principal cell activity

To assess the impact of chemogenetic IN suppression on context-dependent remapping of hippocampal principal cells, we quantified the correlation of spatial activity profiles (Person's $R$) between the familiar and novel context for each place cell, in line with previous studies[10,108]. We further computed population vectors of all place cells in sessions that had at least 5 place cells in each condition (i.e. during baseline and during clozapine) by stacking their average activities over distance on top of each other. Population vector correlations were then determined as the cross-correlation of these cellular activities in each 10-cm bin along the linear track between the familiar and novel context (Supplementary Fig. 7d).

### Statistics

All statistical tests are described in the corresponding figure legends. All comparisons were two-sided. Unless indicated otherwise, statistical comparisons were made between cells fulfilling the individual criteria as specified in the figure legend. The reported $n$ numbers exclude missing ('NaN') values. Throughout the manuscript, extreme outliers were defined as values deviating from their respective empirical distributions by more than 3 interquartile ranges[109] and removed before plotting the respective distributions or performing statistical analysis. The empirical distributions of data for each assessed variable were then tested for normality using the Kolgorov-Smirnov test at an alpha threshold of 0.05. Parametric (Student's $t$ test, ANOVA) or non-parametric tests (Wilcoxon Rank-Sum test, Kruskal–Wallis analysis of variance test) were used as appropriate for normally or non-normally distributed data, respectively. An extensive list of additional statistical comparisons between data and exact $p$ values for all statistical comparisons throughout the manuscript can be found in Supplementary Table 2.

### Reporting summary

Further information on research design is available in the Nature Portfolio Reporting Summary linked to this article.

## Data availability

The processed data for all figures in this manuscript are available through the corresponding Source Data file. The raw dataset on which they are based is available from the corresponding authors upon request. Source data are provided with this paper.

## Code availability

All custom written code for this manuscript has been deposited on github (https://github.com/ThomasHainmueller/HainmuellerCazala_et_al_2023.git).

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

## Acknowledgements

We thank K. Winterhalter and K. Semmler for technical support. We further thank Drs. Claudio Elgueta and Antonio Fernández-Ruiz for comments on earlier versions of this manuscript. This work was funded by the German Research Foundation (DFG CRC/TRR384 M.B.; BA1582/12-1 M.B.; FOR2143 M.B.; HA 8939/1-1 T.H.), EMBO (ALTF 6-2019 T.H.) and by the ERC-AdG 787450 (M.B.).

## Author contributions

T.H. and M.B. conceived the study and designed experiments; T.H., A.C., and L-W.H performed experiments and analyzed data. A.C. performed DREADD experiments. T.H., A.C. and M.B. discussed the data and wrote the manuscript.

## Funding

## Competing interests

The authors declare no competing interests.
