## [Peer Review File · Nature Communications]

Subfield-specific interneuron circuits govern the hippocampal response to novelty in male miceEditorial Note: This manuscript has been previously reviewed at another journal that is not operating a transparent peer review scheme. This document only contains reviewer comments and rebuttal letters for versions considered at *Nature Communications*. Mentions of the other journal have been redacted.

REVIEWERS' COMMENTS

Reviewer #1 (Remarks to the Author):

The authors have addressed all of my concerns. I support publication.

Reviewer #2 (Remarks to the Author):

This manuscript is significantly improved in terms of content from the initial submission to [redacted]. The authors have addressed most of my previous comments by adding new statistical tests and new data.

1) Unfortunately, the new additions have made an already complex and dense paper even more difficult to read and parse out the main points and allow the reader to draw conclusions. The authors do a commendable job of laying out the data and statistical tests in detail for readers to see and evaluate. However, it takes a lot of effort to read this paper, as the reader is forced to bounce from main text figures to supplementary figures to supplementary tables to gather all of the information required to evaluate the robustness and statistical significance of many of the results. I encourage the authors to think about how they can simplify presentation so that the main information needed for readers to evaluate the results is in the main text and figures, with the supplementary figures reserved for supporting data rather than critical information needed to evaluate the significance of the main results. For example, the direct comparison tests requested in the initial reviews are necessary to evaluate some of the claims made, yet the data that show these comparisons (both those that show positive and negative results) are often placed in the supplementary material and only briefly mentioned in the main text. The information is all there, but it is hard to access, and I think few readers will have the patience and time to dig it all out. Similar comments are applicable to the control data of Supplementary Figure 7. I leave it to the authors and editors to decide how, or whether, to address this issue, as it is a question of readability of the paper rather than of scientific validity or impact.

2) I remain unconvinced about the authors' arguments about the magnitude of many of the effects (e.g., that limitations of the technique "may" underestimate the true effect sizes is not a strong argument; that they see a strong effect size in their validation that the DREADDs are actually silencing the neurons in the

intended fashion does not address the question of whether the scientific questions under study here show a strong enough effect to be scientifically meaningful, etc.) and whether their conclusions about different functions played by PV and SOM neurons in different regions are supported strongly and cleanly by these relative weak effects. However, for the intended specialist audience of Nature Communications this is less of a concern, as the data are available for readers to draw their own conclusions. I do wonder whether the authors' choice of violin plots, with connecting lines between individual neurons in the background, contributes to the impression of small magnitudes of effects. The problem is that most of the lines are obscured by the lines occluding each other. I wonder if scatter plots, such as in Figure 4A, are overall a more effective way of portraying the data than the violin plots. For example, seeing that almost all of the points are above the diagonal for DG-SOM in Fig 4A is visually more compelling than seeing the small shift in the medians and the highly overlapping distributions in the violin plots of Fig 4A. I leave it to the authors to decide if they believe that their message will be more effectively conveyed by presenting more scatter plots in place of (or in addition to) the violin plots.

3) For future work (not in the present paper), I recommend that the authors consider the use of mixed-effects models to account for the contributions of individual animals and cells within an animal, utilizing these variables as random effects in the models. These techniques are becoming increasingly used in neuroscience to account for the problems that arise by questions of whether the individual cell, the individual session, or the individual animal are the proper units of measurement for statistical significance testing. This is especially problematic as the number of simultaneously recorded neurons, from both imaging and high-density electrode arrays, increases dramatically with new technologies and magnifies this problem. The use of mixed effects models potentially eliminates the awkwardness and ambiguity of showing results that are highly significant when using hundreds of cells that are not independent samples, but having the significance disappear when using individual animals or sessions as the unit of analysis.

Reviewer #3 (Remarks to the Author):

The authors have done a superb job at addressing my concerns.

REVIEWERS' COMMENTS

Reviewer #1 (Remarks to the Author):

The authors have addressed all of my concerns. I support publication.

Reviewer #2 (Remarks to the Author):

This manuscript is significantly improved in terms of content from the initial submission to [redacted]. The authors have addressed most of my previous comments by adding new statistical tests and new data.

1) Unfortunately, the new additions have made an already complex and dense paper even more difficult to read and parse out the main points and allow the reader to draw conclusions. The authors do a commendable job of laying out the data and statistical tests in detail for readers to see and evaluate. However, it takes a lot of effort to read this paper, as the reader is forced to bounce from main text figures to supplementary figures to supplementary tables to gather all of the information required to evaluate the robustness and statistical significance of many of the results. I encourage the authors to think about how they can simplify presentation so that the main information needed for readers to evaluate the results is in the main text and figures, with the supplementary figures reserved for supporting data rather than critical information needed to evaluate the significance of the main results.

For example, the direct comparison tests requested in the initial reviews are necessary to evaluate some of the claims made, yet the data that show these comparisons (both those that show positive and negative results) are often placed in the supplementary material and only briefly mentioned in the main text. The information is all there, but it is hard to access, and I think few readers will have the patience and time to dig it all out. Similar comments are applicable to the control data of Supplementary Figure 7. I leave it to the authors and editors to decide how, or whether, to address this issue, as it is a question of readability of the paper rather than of scientific validity or impact.

We thank the referee for his/her/their suggestions and made a sincere effort to improve the legibility and arrangement of the data in our paper. We followed the referee's suggestions and now show the direct comparison of the groups from former Supplementary figure 7k,l on our revised main Figure 6l,m. We further show the control data from former Supplementary figure 7i and j on the new main Figures 6i-k and 7e,f, respectively. We further show the data for the interneuron suppression mentioned below by the referee, which we think is critical for the evaluation of our data, on the new main figure 5 (formerly on supplementary Fig 7a-d). We are convinced that this rearrangement improves the accessibility of our most important findings.

2) I remain unconvinced about the authors' arguments about the magnitude of many of the effects (e.g., that limitations of the technique "may" underestimate the true effect sizes is not a strong argument; that they see a strong effect size in their validation that the DREADDs are actually silencing the neurons in the intended fashion does not address the question of whether the scientific questions under study here show a strong enough effect to be scientifically meaningful,

etc.) and whether their conclusions about different functions played by PV and SOM neurons in different regions are supported strongly and cleanly by these relative weak effects. However, for the intended specialist audience of Nature Communications this is less of a concern, as the data are available for readers to draw their own conclusions.

I do wonder whether the authors' choice of violin plots, with connecting lines between individual neurons in the background, contributes to the impression of small magnitudes of effects. The problem is that most of the lines are obscured by the lines occluding each other. I wonder if scatter plots, such as in Figure 4A, are overall a more effective way of portraying the data than the violin plots. For example, seeing that almost all of the points are above the diagonal for DG-SOM in Fig 4A is visually more compelling than seeing the small shift in the medians and the highly overlapping distributions in the violin plots of Fig 4A. I leave it to the authors to decide if they believe that their message will be more effectively conveyed by presenting more scatter plots in place of (or in addition to) the violin plots.

We thank the reviewer for this suggestion and added more scatterplots (e.g. revised figure 4c and d). While we generally agree that scatterplots are a useful alternative to display the data, the manuscript at this point shows so many comparisons (e.g. Figure 3 and 6 alone would require altogether 18 or 12 scatterplots, respectively, to convey the same information) that we don't think it is feasible to replace the violin plots throughout.

3) For future work (not in the present paper), I recommend that the authors consider the use of mixed-effects models to account for the contributions of individual animals and cells within an animal, utilizing these variables as random effects in the models. These techniques are becoming increasingly used in neuroscience to account for the problems that arise by questions of whether the individual cell, the individual session, or the individual animal are the proper units of measurement for statistical significance testing. This is especially problematic as the number of simultaneously recorded neurons, from both imaging and high-density electrode arrays, increases dramatically with new technologies and magnifies this problem. The use of mixed effects models potentially eliminates the awkwardness and ambiguity of showing results that are highly significant when using hundreds of cells that are not independent samples, but having the significance disappear when using individual animals or sessions as the unit of analysis.

We thank the referee for this kind suggestion for our future work, which we are happy to consider in our next projects.

Reviewer #3 (Remarks to the Author):

The authors have done a superb job at addressing my concerns.